# COVID-19 vaccine acceptance, hesitancy, and associated factors among medical students in Sudan

Saud Mohammed Raja[1,2]*, Murwan Eissa Osman[1], Abdelmageed Osman Musa[1], Asim Abdelmoneim Hussien[1], Kabirat Yusuf[1]

1 Department of Community Medicine, Faculty of Medicine, International University of Africa, Khartoum, Sudan, 2 Department of Internal Medicine, Sudan Medical Specialization Board, Khartoum, Sudan

* dr.saudraja@gmail.com

## Abstract

### Background

The COVID-19 vaccination in Sudan launched in March 2021 but the extent of its acceptance has not been formally studied. This study aimed to determine the acceptance and hesitancy of the COVID-19 vaccine and associated factors among medical students in Sudan.

### Methods

A descriptive cross-sectional study was conducted using an online self-administered questionnaire designed on Google Form and sent to randomly-selected medical students via their Telegram accounts from 30th June to 11th July 2021. Data were analyzed using Statistical Package for Social Sciences software. Chi-square or Fisher's exact test and logistic regression were used to assess the association between vaccine acceptance and demographic as well as non-demographic factors.

### Results

Out of the 281 students who received the questionnaire, 220 (78%) responded, of whom 217 consented and completed the form. Males accounted for 46. 1%. Vaccine acceptance was 55. 8% (n = 121), and vaccine hesitancy was 44. 2% (n = 96). The commonly cited reasons for accepting the vaccine were to protect themselves and others from getting COVID-19. Concerns about vaccine safety and effectiveness were the main reasons reported by those who were hesitant. Factors associated with vaccine acceptance were history of COVID-19 infection (adjusted odds ratio (aOR) = 2. 2, 95% CI 1. 0–4.7, $p$ = 0. 040), belief that vaccines are generally safe (aOR = 2.3, 95% CI 1. 2–4.5, $p$ = 0.020), confidence that the vaccine can end the pandemic (aOR = 7.5, 95% CI 2. 5–22. 0, $p$<0.001), and receiving any vaccine in the past 5 years (aOR = 2.4, 95% CI 1.1–5.4, $p$ = 0.031). No demographic association was found with the acceptance of the vaccine.

**Data Availability Statement:** All relevant data are within the manuscript and its Supporting Information files.

**Funding:** The authors received no specific funding for this work.

**Competing interests:** The authors have declared that no competing interests exist.

## Conclusions

This study has revealed a high level of COVID-19 vaccine hesitancy among medical students. Efforts to provide accurate information on COVID-19 vaccine safety and effectiveness are highly recommended.

## Introduction

More than a year since its declaration of Coronavirus Disease 2019 (COVID-19) as a pandemic on 11 March 2020 [1], the disease is still causing enormous public health crisis globally with ongoing infections, mortality, and serious economic and social impact [2]. As of August 2021, Over 217 million people worldwide have been infected with the culprit virus named severe acute respiratory syndrome-2 (SARS-CoV-2) resulting in over 4. 5 million deaths [3]. Sadly, the COVID-19 pandemic is expected to keep on imposing huge morbidity and mortality and disrupt societies and economies globally [4].

Although several preventive and therapeutic measures have been attempted, the newly-developed vaccines have also been shown to be effective at minimizing the effect of COVID-19 [5]. Indeed, public health measures such as quarantine, social distancing, mask-wearing, and other non-pharmaceutical interventions played an important role in mitigating the spread of the pandemic [6]. Nevertheless, the rate of new infections and deaths is still on the rise. Several vaccines have been developed with some already authorized and others under clinical trials. Notably, the Pfizer/BioNtech, AstraZeneca, Moderna, and Johnson & Johnson's Janssen COVID-19 vaccines are among the popular preparations approved for emergency use in several countries [7]. Regardless of the disproportionate distribution of the vaccines between high-income and low-income countries, vaccine hesitancy remains a significant barrier to the successful inoculation of the public [8].

According to the World Health Organization (WHO), vaccine hesitancy is defined as reluctance or refusal to vaccinate despite the availability of vaccines [9]. The WHO designated vaccine hesitancy as one of the ten threats to global health in 2019 [10]. Surprisingly, in the era of COVID-19, the problem is significant even among healthcare workers [11]. Medical students are future healthcare professionals who are considered open-minded and expected to respond quickly to public health measures. Unfortunately, a study conducted in the USA reported that nearly a quarter of the medical students were unwilling to get the COVID-19 vaccine as soon as the vaccines are approved [12]. Another study in India also showed that 10. 6% of medical students were hesitant to take the vaccine [13]. It has been reported that at least one in five healthcare trainees, on average, is hesitant for the COVID-19 vaccine [14]. In Africa, despite the scarcity of the vaccine itself, acceptability for it seems lower than in the developed world. A nationwide survey conducted among medical students in Uganda reported that more than a third of them were hesitant for the COVID-19 vaccine [15].

Several factors are associated with the acceptance of medical students for the vaccine. Importantly, the source of information plays a pivotal role [15]. The role of social media in spreading negative information about COVID-19 vaccination is evident. Not only is the pandemic at a continuous spread despite all the measures, disinformation and misinformation about the infection and the vaccine are also spreading at a similar pace, if not faster. The situation, known as "infodemic", has a strong impact on public health response to the COVID-19 pandemic [16]. Noticeably, medical students are expected to be influenced significantly as they are the age group where social media is pervasive.

COVID-19 has been reported in Sudan since March 13, 2020. To date, more than thirty-seven thousand people have been infected with over 2,800 deaths [17]. Nearly a year after the first case, Sudan received over 800,000 doses of AstraZeneca vaccine on March 3, 2021, through COVID-19 Vaccines Global Access (COVAX) Facility [18] and vaccinations started on March 9 prioritizing healthcare workers, the aged population, and those with chronic medical conditions. Two shots of the vaccine have been delivered thereby allowing some people to get fully vaccinated. However, many medical professionals seem to be less enthusiastic about the vaccination roll-out.

To the best of our knowledge, no study has been conducted in Sudan to address COVID-19 vaccine acceptance among medical students. Notably, medical students are an easily targetable population to be good role models in the community and foster positive public health opinions. Hence, we aimed to assess the acceptance and hesitancy for the COVID-19 vaccine and associated factors among medical students in Sudan.

## Methods

### Study design

We conducted an online, descriptive, cross-sectional, institution-based study.

### Study area

The study was carried out at the International University of Africa (IUA), Faculty of Medicine. Based in Khartoum, Sudan, the faculty aims to qualify medical doctors for the diagnosis and treatment of endemic diseases and other health problems [19] who can fill the gap in the health workforce crisis. The faculty can accommodate more than a thousand medical students.

### Study population

Fourth and fifth-year (clinical phase) medical students at the IUA, Faculty of Medicine were selected as the study population. At the time of the study, 471 medical students were enrolled in the two clinical phase years from which the sample population was recruited.

### Inclusion and exclusion criteria

Medical students enrolled and active in their clinical year studies during the study period who joined the main social media group (Telegram) and consented to the study were included. Those who lacked access to the Internet were excluded.

### Variables

Dependent variable: COVID-19 vaccine acceptance, hesitancy.

Independent variables: COVID-19 vaccine safety, efficacy, perceived risk of COVID-19 infection or pandemic, prior COVID-19 infection, previous vaccination, confidence in the vaccine, source of information.

Background variables: Age, gender, marital status, year of study, nationality.

### Sampling

**Sample size.** The Miller and Brewer's mathematical formula [20] was used to calculate the sample size from the total 471 medical students.

$n = N/[1+N(d)^2]$

*Where, N = the population size (471)*

$d$ = the degree of accuracy required (0.05)

n = 471/1+[471 (0.05)$^2$]

n = 216 (before adjusting for non-response)

To adjust for the non-response rate, which was estimated to be up to 30%, the sample size was augmented as follows.

*Sample size for data collection* = 216 + (0.3 x 216) = 281

After data collection, the final sample size that fulfilled the inclusion criteria and thus included in data analysis was 217.

**Sampling technique.** A simple random sampling technique was employed to recruit participants into the study. A list of the students (sampling frame) was obtained and randomly selected individuals were contacted.

## Data collection methods

A validated questionnaire with closed-ended questions from previous studies by Kanyike et al [15] as well as El-Elimat and colleagues [21] was adapted and modified to suit the study participants. The questionnaire was developed in Google Form and a link of it was sent individually to randomly selected participants into their Telegram account obtained from the Telegram group of the students.

To minimize the non-response rate, the coordinator for each group initially shared an announcement in the main group so that randomly selected students expect a message in their personal accounts. Data collection started on 30th June 2021 and ended on 11th July 2021. Two reminder messages more than 48 hours apart were sent both in the main group and to the Telegram account of the individual participants who didn't respond quickly. When almost 90% of the sample size was reached and the response rate was minimal, a 24-hour deadline reminder was sent, after which receiving responses was terminated.

## Data analysis

The fully completed forms were exported to Microsoft Excel for cleaning and coding. Cleaned data were fed into Statistical Package for Social Sciences program (SPSS) Version 23 for analysis. The categorical data were summarized frequencies and percentages. Associations between variables were assessed using the Chi-square test or Fischer's exact test. The WHO definition of vaccine hesitancy [22] was adopted to categorize participants into those who accept the COVID-19 vaccine and hesitant ones. Binary logistic regression was applied for variables that showed statistically significant correlation to identify statistically significant predictor variables. A $p$-value $<0.05$ was considered statistically significant. Results were presented with graphs, charts, and tables as appropriate.

## Ethics approval and consent to participate

The study was approved by an ethical clearance obtained from the IUA, Faculty of Medicine, and Deanship of Higher Education, Research, and Publications. Only adult participants voluntarily willing to take part in the study were included by accepting an electronic written informed consent at the initial page of the online questionnaire. The ethical principles related to the inclusion of human subjects were strictly followed as defined in the Nuremberg Code and the Declaration of Helsinki.

## Results

### Demographics

Out of the 281 randomly selected students who received the questionnaire, 220 (78%) responded, of whom 217 consented and completed the questionnaire. Three-quarters of the students were below or at the age of 24 years, with females slightly more than males, and nearly all students were single. More than two-thirds of the students were non-Sudanese. Table 1 summarizes the demographic characteristics of the participants.

### COVID-19 vaccine acceptance and hesitancy

The majority of the students (n = 121, 55.8%) accepted the vaccine. The most commonly reported reason for the acceptance of the vaccine was to protect themselves from getting the vaccine (nearly 80%), followed by both to protect others and to be able to travel. On the other hand, university recommendation was the least mentioned reason for taking the COVID-19 vaccine.

Ninety-six (44.2%) students were not willing to take the COVID-19 vaccine, of whom 71 (32.7%) were not sure whether they would decide to take it and 25 (11.5%) reported that they made up their mind will not take the vaccine. Most participants were hesitant about the vaccination due to concerns related to the safety and effectiveness of the vaccine as well as the influence of negative information about the COVID-19 vaccine. Six participants clearly reported that they were against any form of vaccination. Table 2 summarizes the reasons for acceptance and hesitancy of the COVID-19 vaccine among the medical students.

### Factors associated with COVID-19 vaccine acceptance and hesitancy

Demographic factors, perceived risk of COVID-19 infection personally or to the public, the safety of vaccines in general, history of COVID-19 infection, and sources of information were assessed for association with acceptance or hesitancy of the COVID-19 vaccine among the students. As shown in Table 3, none of the demographic characteristics of the participants had a statistically significant correlation with vaccine acceptance and hesitancy. However, several

**Table 1. Demographic characteristics of medical students at IUA, Khartoum, Sudan, 2021 (n = 217).**

| Demographics | Frequency | Percentage |
|---|---|---|
| **Age** | | |
| ≤ 24 | 161 | 74.2 |
| > 24 | 56 | 25.8 |
| **Sex** | | |
| Male | 100 | 46.1 |
| Female | 117 | 53.9 |
| **Marital Status** | | |
| Single | 209 | 96.3 |
| Married | 6 | 2.8 |
| Prefer not to say | 2 | 0.9 |
| **Year of study** | | |
| First clinical year (Year 4) | 99 | 45.6 |
| Second clinical year (Year 5) | 118 | 54.4 |
| **Nationality** | | |
| Sudanese | 62 | 28.6 |
| Non-Sudanese | 155 | 71.4 |

**Table 2. Reasons for acceptance and hesitancy of the COVID-19 vaccine among medical students at IUA, Khartoum, Sudan, 2021 (n = 121).**

| Reason | Frequency | Percentage |
|---|---|---|
| **Reasons for accepting (n = 121)** | | |
| To protect myself from getting COVID-19 | 96 | 79.3 |
| To protect others from getting COVID-19 | 71 | 58.7 |
| To be able to travel | 71 | 58.7 |
| I believe in vaccines and immunization | 56 | 46.3 |
| I believe the vaccines are effective | 47 | 38.8 |
| It is a social and moral responsibility | 43 | 35.5 |
| To get rid of the virus and end the pandemic | 41 | 33.9 |
| Health workers' recommendations | 36 | 29.8 |
| I believe the vaccines are safe | 35 | 28.9 |
| Government recommendations | 19 | 15.7 |
| Parent recommendations | 18 | 14.9 |
| I am at high risk of severe disease | 5 | 4.1 |
| University recommendations | 2 | 1.7 |
| **Reasons for hesitancy (n = 91)** | | |
| I am concerned of the vaccine safety | 38 | 39.6 |
| I am not sure of the vaccine effectiveness | 35 | 36.5 |
| I have heard or read negative information about the vaccines | 34 | 35.4 |
| I want to wait until more people take it | 18 | 18.8 |
| I am young. I can recover easily if infected | 13 | 13.5 |
| I do not believe it is important | 13 | 13.5 |
| Vaccine development was rushed | 13 | 13.5 |
| Someone I know had bad reaction after vaccination | 11 | 11.5 |
| Someone else told me that the vaccine was not safe | 10 | 10.4 |
| I feel I had enough immunity | 10 | 10.4 |
| I am busy with my studies | 9 | 9.4 |
| I don't know why | 9 | 9.4 |
| I had or maybe had COVID-19 infection already | 8 | 8.3 |
| I am against any form of vaccination | 6 | 6.3 |
| The vaccine brand I prefer was not available locally | 5 | 5.2 |
| I don't know where to get vaccinated | 4 | 4.2 |
| I fear needles | 4 | 4.2 |

other factors were found to be associated. Acceptance and hesitancy for the COVID-19 vaccine among the participants were significantly associated with consideration of COVID-19 as a public health threat, worry about the infection, belief of being the pandemic a threat to people in Sudan, the attitude that vaccines are generally safe, the trust that COVID-19 vaccination may end the pandxxxxxemic, and having heard negative information about the COVID-19 vaccine. Table 4 summarizes the non-demographic associated factors and Fig 1 illustrates the source of information regarding the COVID-19 pandemic and its vaccine among the participants.

## Predictors associated with vaccine acceptance among medical students

Table 5 shows how predictive the associated factors are for COVID-19 vaccine acceptance among the medical students when the logistic regression model was applied. Variables that were found to be predictive of the COVID-19 vaccine were history of COVID-19 infection,

**Table 3. Association of demographic characteristics with COVID-19 vaccine acceptance and hesitancy among medical students at IUA, Khartoum, Sudan, 2021 (n = 217).**

| Variables | Acceptance of COVID-19 vaccine | | | | Chi-Square | p-value |
|---|---|---|---|---|---|---|
| | Yes *(n = 121)* | | No *(n = 96)* | | | |
| | *frequency* | *%* | *frequency* | *%* | *Chi-Square* | *p-value* |
| **Age** | | | | | | |
| ≤ 24 | 92 | 57. 1 | 69 | 42. 9 | 0. 483 | 0. 487 |
| > 24 | 29 | 51. 9 | 27 | 48. 1 | | |
| **Sex** | | | | | | |
| Male | 57 | 57 | 43 | 43 | 0. 115 | 0. 734 |
| Female | 64 | 54. 7 | 53 | 45. 3 | | |
| **Marital Status** | | | | | | |
| Single | 115 | 55. 0 | 94 | 45. 0 | 1. 990* | 0. 437 |
| Married | 5 | 83. 3 | 1 | 16. 7 | | |
| Prefer not to say | 1 | 50. 0 | 1 | 50. 0 | | |
| **Year of study** | | | | | | |
| Year 4 | 54 | 54. 5 | 45 | 45. 5 | 0. 109 | 0. 741 |
| Year 5 | 67 | 56. 8 | 51 | 43. 2 | | |
| **Nationality** | | | | | | |
| Sudanese | 37 | 59. 7 | 25 | 40. 3 | 0. 534 | 0. 462 |
| Non-Sudanese | 84 | 54. 2 | 71 | 45. 8 | | |

* Fisher's exact test was done instead of Chi-square test.

belief in the general safety of vaccines, the trust that the COVID-19 vaccine may end the pandemic, and vaccination for other diseases in the last five years. Notably, those who think the COVID-19 vaccine can end the pandemic if enough people in the world get vaccinated are more than seven times more likely to accept the COVID-19 vaccine than those who refute the possibility.

## Discussion

The WHO has earlier recognized vaccine hesitancy as one of the global health threats [10] and in the era of the COVID-19 pandemic, the issue has become of more concern than ever before [8]. Of note, COVID-19 vaccine hesitancy among healthcare workers, including medical students, has been recently reported to be a significant trend. This study aimed to find out the COVID-19 vaccine acceptance and hesitancy as well as associated factors among medical students at one of the leading medical schools in Sudan. To the best of our knowledge, this is the first study to address the situation among healthcare students in Sudan.

This study revealed that 55. 8% of the medical students accepted the COVID-19 vaccine. This acceptance level is higher than reported among medical students in Uganda (37. 3%) [15] and Egypt (35%) [23]. The comparatively higher acceptance in this study could be due to the gradually increasing knowledge and trust of the vaccine unlike at the beginning of the vaccine rollout when the former studies were conducted. In contrast, the finding in this study was much lower compared to the acceptance rates among medical students in India (89. 4%) [13] and Poland (95. 9%) [24] as well as university students in Italy (94. 7%) [25]. The disparity in the burden of the COVID-19 pandemic in regions such as India and Europe compared to Africa could presumably justify the exceedingly higher acceptance in those highly affected areas than found in this study. The main reasons cited for accepting the vaccine were personal

**Table 4. Non-demographic factors associated with COVID-19 vaccine acceptance and hesitancy among medical students at IUA, Khartoum, Sudan, 2021 (n = 217).**

| Variables | Acceptance of COVID-19 vaccine | | | | Chi-Square* | p-value |
|---|---|---|---|---|---|---|
| | Yes (n = 121) | | No (n = 96) | | | |
| | frequency | % | frequency | % | | |
| **Believes COVID-19 is a real public health threat at present.** | | | | | | |
| Yes | 99 | 61. 9 | 61 | 38. 1 | 10. 806* | 0. 010** |
| No | 2 | 20. 0 | 8 | 80. 0 | | |
| May be | 17 | 42. 5 | 23 | 57. 5 | | |
| Don't know | 3 | 42. 9 | 4 | 57. 1 | | |
| **Worried about COVID-19 infection.** | | | | | | |
| Not at all worried | 17 | 40. 5 | 25 | 59. 5 | | |
| Somewhat worried | 30 | 68. 2 | 14 | 31. 8 | | |
| Not very worried | 44 | 53. 7 | 38 | 46. 3 | 9. 563* | 0. 046** |
| Very worried | 27 | 65. 9 | 14 | 34. 1 | | |
| Extremely worried | 3 | 37. 5 | 5 | 62. 5 | | |
| **COVID-19 poses a risk to me personally.** | | | | | | |
| No risk at all | 12 | 46. 2 | 14 | 53. 8 | | |
| Minor risk | 29 | 51. 8 | 27 | 48. 2 | | |
| Moderate risk | 43 | 60. 6 | 28 | 39. 4 | 4. 507 | 0. 342 |
| Major risk | 21 | 67. 7 | 10 | 32. 3 | | |
| Don't know | 16 | 48. 5 | 17 | 51. 5 | | |
| **COVID-19 poses a risk to people in Sudan.** | | | | | | |
| No risk at all | 5 | 83. 5 | 1 | 16. 7 | 12. 938* | 0. 010** |
| Minor risk | 11 | 40. 7 | 16 | 59. 3 | | |
| Moderate risk | 33 | 45. 2 | 40 | 54. 8 | | |
| Major risk | 55 | 68. 8 | 25 | 31. 3 | | |
| I don't know | 17 | 54. 8 | 14 | 45. 2 | | |
| **Estimated likelihood of getting COVID-19 infection in the future.** | | | | | | |
| Not at all | 16 | 40. 0 | 23 | 59. 0 | 4. 727* | 0. 317 |
| Slightly | 45 | 57. 0 | 34 | 43. 0 | | |
| Moderate | 39 | 60. 9 | 23 | 39. 1 | | |
| Very likely | 18 | 58. 1 | 13 | 41. 9 | | |
| Extremely likely | 3 | 75. 0 | 1 | 25. 0 | | |
| **History of COVID-19 infection.** | | | | | | |
| No | 57 | 52. 8 | 51 | 47. 2 | 5. 731 | 0. 125 |
| Yes, confirmed | 13 | 72. 2 | 5 | 27. 8 | | |
| Yes, not confirmed | 29 | 65. 9 | 15 | 34. 1 | | |
| Not sure | 22 | 46. 8 | 25 | 53. 2 | | |
| **In general, vaccines are safe.** | | | | | | |
| Disagree | 0 | - | 14 | 100. 0 | 23. 595 | 0. 000** |
| Neutral | 24 | 47. 1 | 27 | 52. 9 | | |
| Agree | 77 | 62. 6 | 46 | 37. 4 | | |
| Strongly agree | 20 | 69. 0 | 9 | 31. 0 | | |
| **COVID-19 vaccine will end the pandemic if enough people in the world get it.** | | | | | | |
| Strongly disagree | 2 | 50. 0 | 2 | 50. 0 | 41. 457* | 0. 000** |
| Disagree | 5 | 21. 7 | 18 | 78. 3 | | |
| Neutral | 24 | 35. 3 | 44 | 64. 7 | | |
| Agree | 66 | 70. 2 | 28 | 29. 8 | | |
| Strongly agree | 24 | 85. 7 | 4 | 14. 3 | | |

*(Continued)*

**Table 4.** (Continued)

| Variables | Acceptance of COVID-19 vaccine | | | | | |
|---|---|---|---|---|---|---|
| | Yes (n = 121) | | No (n = 96) | | | |
| | *frequency* | *%* | *frequency* | *%* | *Chi-Square** | *p-value* |
| **History of vaccination for other diseases in the last 5 years.** | | | | | | |
| Not vaccinated | 14 | 41. 2 | 20 | 58. 8 | 3. 573* | 0. 139 |
| Vaccinated | 103 | 58. 9 | 72 | 41. 1 | | |
| Don't remember | 4 | 50. 0 | 4 | 50. 0 | | |
| **Heard any negative information about the COVID-19 vaccine.** | | | | | | |
| No | 11 | 73. 3 | 4 | 26. 7 | | 0. 047** |
| Yes | 106 | 56. 4 | 82 | 43. 6 | 6. 103 | |
| Not sure | 4 | 28. 6 | 10 | 71. 4 | | |
| **Use of social media as an information source for COVID-19** | | | | | | |
| Yes | 69 | 56. 1 | 54 | 43. 9 | 0. 013 | 0. 909 |
| No | 52 | 55. 3 | 42 | 44. 7 | | |

* Fisher's exact test was conducted instead of Chi-square test.

** Statistically significant (p value <0. 05)

and others' protection from COVID-19 infection, a finding similar to the Ugandan, Egyptian, and Polish studies [15, 23, 24]. Furthermore, more than half of the participants in this study reported accepting the vaccine for travel reasons, which is increasingly becoming important as "COVID-19 vaccine passports" are being introduced [26].

COVID-19 vaccine hesitancy among the medical students in this study was found to be 44. 2%. This rate of hesitancy is comparable with that reported in Egypt (47. 1%) [23] and among dental students in the USA (44%) [27]. However, the finding was higher in contrast to several

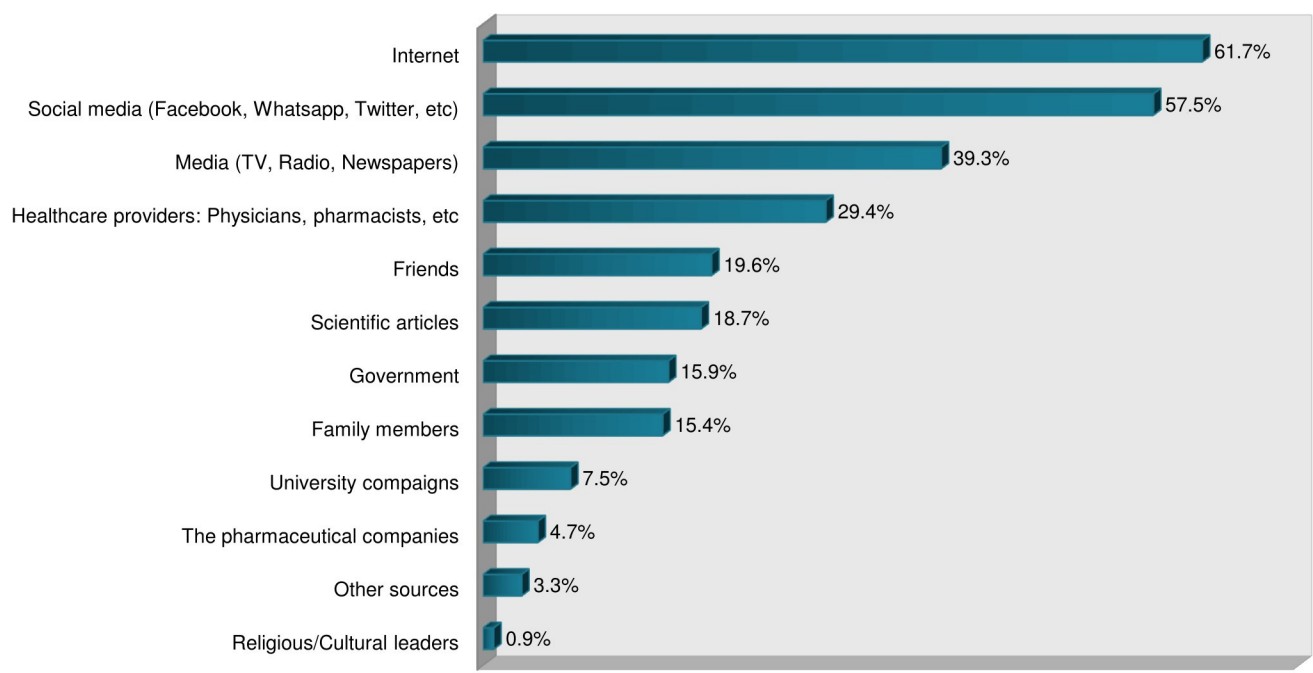

**Fig 1. Sources of information regarding COVID-19 pandemic and vaccines among medical students at IUA, Khartoum, Sudan, 2021 (n = 217).**

**Table 5. Predictors of COVID-19 vaccine acceptance using logistic regression analysis among medical students at IUA, Khartoum, Sudan, 2021 (n = 217).**

| Predictors variable* | Adjusted odds ratio (aOR) | 95% CI | *p*-value |
|---|---|---|---|
| **Believes that COVID-19 is a real public health threat at present.** | | | |
| No/Maybe/Don't know | Reference | | |
| Yes | 1. 7 | 0. 8–3. 5 | 0. 175 |
| **Worried about COVID-19 infection.** | | | |
| Not worried | Reference | | |
| Worried | 2. 1 | 0. 9–4. 9 | 0. 076 |
| **COVID-19 poses a risk to people in Sudan.** | | | |
| No/Minor risk | Reference | | |
| Moderate/Major risk | 0. 9 | 0. 3–2. 2 | 0. 763 |
| Don't know | 1. 0 | 0. 3–3. 5 | 0. 950 |
| **History of previous COVID-19 infection.** | | | |
| No/Not sure | Reference | | |
| Yes | 2. 2 | 1. 0–4. 7 | 0. 040** |
| **In general, vaccines are safe.** | | | |
| Disagree/Neutral | Reference | | |
| Agree/Strongly agree | 2. 3 | 1. 2–4. 5 | 0. 020** |
| **COVID-19 vaccine will end the pandemic if enough people in the world get it.** | | | |
| Disagree/Strongly disagree | Reference | | |
| Neutral | 1. 5 | 0. 5–4. 6 | 0. 476 |
| Agree/Strongly agree | 7. 5 | 2. 5–22. 0 | 0. 000** |
| **History of vaccination for other diseases in the last 5 years.** | | | |
| No/Don't remember | Reference | | |
| Yes | 2. 4 | 1. 1–5. 4 | 0. 031** |
| **Heard any negative information about the COVID-19 vaccine.** | | | |
| No/not sure | Reference | | |
| Yes | 1. 1 | 0. 4–2. 7 | 0. 892 |

* Similar categories of responses of predictor variables are combined to avoid low count cells and make the logistic regression more accurate.

** Statistically significant.

CI = Confidence Interval.

studies conducted on medical students from Poland (4. 1%) [24], India (10. 6%) [13], USA (23%) [12], and the overall rate among healthcare students or workers from 39 countries (18. 9%) [14]. These discrepancies could be explained by the difference in the time of conducting the study, the disparity in the burden of COVID-19infection across the world, inconsistency in the exact definition of "vaccine hesitancy", and presumably the variable efforts of responsible bodies to minimize vaccine hesitancy both in the healthcare professionals and the public. Although this study was conducted among medical students in Sudan, more than two-thirds of the students at IUA come from various countries, mainly from neighboring African and Arab countries.

The two main reasons for the COVID-19 vaccine hesitancy among the medical students in this study were concerns related to vaccine safety and effectiveness, as shown in Table 2. This is consistent with the bulk of the literature as reported by similar studies from Egypt [23], Uganda [15], India [13],USA [12],Turkey [28], Poland [29], Slovenia [30],and China [31] among others. More than a third of the COVID-19 vaccine-hesitant group also stated negative

information about the vaccine as one of their main causes for deferring vaccination, in congruence with the study by Kanyike et al. [15]. Similarly, having heard or read negative information about the COVID-19 vaccines was also among the top grounds reported by the medical students for being hesitant regarding the vaccination.

To date, no published report exists regarding the extent of COVID-19 vaccine hesitancy among the general population in Sudan. However, based on our findings, we can presume a similar or higher rate of hesitancy in the public, which needs to be confirmed in a further study. Apart from the vaccine safety and effectiveness concerns, the medical students in our study expressed their lack of confidence in the vaccine, and thus deferral of the vaccination, in various ways. Nearly one in five opted to wait and see until others take the vaccine. One in ten was influenced by someone who told them that the vaccine was unsafe or had suffered from an adverse reaction. Some participants believed the vaccine was unnecessary with a minority who showed the attitude of rejecting the concept of vaccination at all. A significant number of medical students thought being young and healthy would protect them from the pandemic. Of course, being busy students of medicine, lack of awareness of the local vaccination programs, and absence of access to the vaccine within reach kept certain participants hesitant. It is worth mentioning that some participants reported personal reasons for hesitancy that fitted one of the listed reasons in Table 2.

We also addressed the factors associated with COVID-19 vaccine acceptance and hesitancy among the clinical phase students. In contrast to some studies [15, 23, 32] that have determined a significant association of demographic factors such as gender, marital status, or year of study with vaccine acceptance, this study did not find any statistically significant demographic correlation with the decision to accept or defer the COVID-19 vaccine among the medical students. This shows that the medical students were demographically similar without a demographic-based predilection for vaccination. Statistically significant association upon Chi-square or Fisher's exact test were found with the belief of COVID-19 pandemic as a public health threat, being worried of the infection, perceived risk of the pandemic to people in Sudan, attitude to the general safety of vaccines, the role of the vaccine in ending the pandemic, and having heard negative information about the COVID-19 vaccine. These findings were consistent with the similar study conducted in Ugandan medical schools [15]. Indeed, the resemblance of the medical students between Uganda and Sudan and their shared characteristics would justify the comparable finding. After all, a significant proportion of IUA students come from various regions of Africa, thereby strengthening the likeness.

Some associated factors can be utilized to predict COVID-19 vaccine acceptance using a binary logistic regression model. To that ends, upon utilizing the technique, statistically significant predictors of COVID-19 vaccine acceptance in this study were found to be prior infection with COVID-19, previous vaccination for other infections, the belief that vaccines are generally safe, and confidence that COVID-19 vaccination may end the pandemic. Of note, those who agreed that sufficient mass vaccination could end the pandemic were seven times more likely to accept the COVID-19 vaccine when compared to those who refuted the idea. This notion was not generally assessed in previous similar studies. In harmony with the findings of Kanyike et al., [15] this study determined that individuals previously vaccinated for other diseases are more likely to accept the new vaccine than those who didn't.

Contrary to the expectation assumed at the beginning of the study and in opposition to similar studies conducted by Kanyike et al. [15], Sallam et al. [33], and Saied et al. [23], relying on social media as a source of information was not found to be significantly associated with vaccine hesitancy among our medical students. This contradiction may be explained by the fact that the medical students generally use social media for most of the information they get, whether good or bad. In addition, unlike earlier during the pandemic and the rollout of the

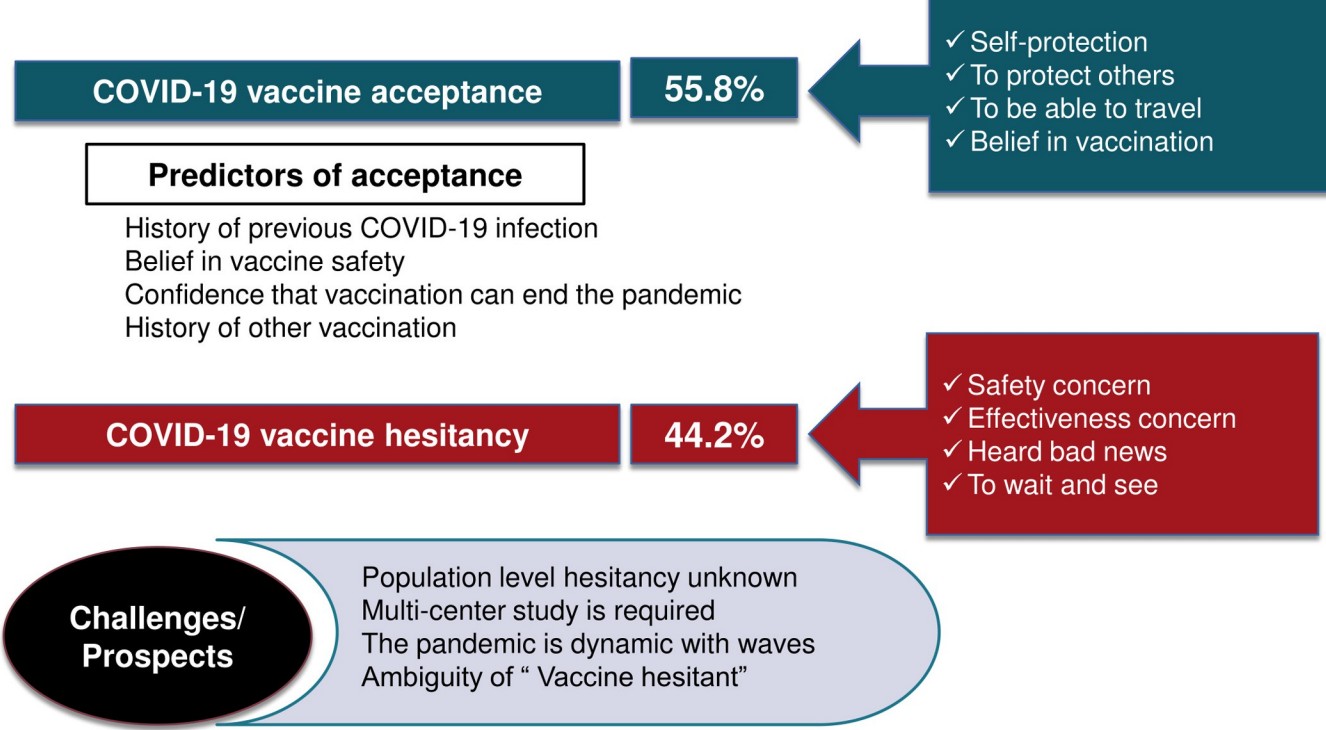

**Fig 2. Summary, challenges, or prospect of the present study.**

vaccine, positive information about successful vaccination against the pandemic is probably being promoted on social media. Nonetheless, medical students are undoubtedly encouraged to rely more on official sources of information and scientific articles in matters like this.

## Limitations of the study

The current study sample was from a single medical school in Sudan. In addition, the majority of the students at IUA are foreigners. Hence, the results from this study may be difficult to project to all medical students in Sudan and may not represent the situation among healthcare professionals or trainees elsewhere in the country. Studying a single out of the numerous medical schools in Khartoum may not give a precise picture of the problem. Definition of vaccine hesitancy according to the WHO was adopted, where indecisive and those who refuse the vaccine are collectively categorized as vaccine-hesitant. However, due to the inconsistency in the literature, some comparisons might be slightly distorted. Since COVID-19 vaccination is not mandatory so far, those who got the vaccine were considered among the accepting group regardless of whether they were hesitant or not prior to vaccination irrespective of some strict technical definitions that may be contrary to the operational definition adopted in this study. Finally, knowledge of the students on COVID-19 and how it affects their decision to accept the vaccine was not assessed to keep the questionnaire a reasonable length for an adequate response. A visual summary of the main findings and the challenges or prospects of this study is provided in Fig 2.

## Strengths of the study

We applied a random sampling technique to recruit participants, which was barely applied in other studies of the COVID-19 vaccine acceptance survey among healthcare personnel or trainees. On top of that, nearly 80% response rate was a huge achievement compared to several similar studies. This was obtained by choosing the most conducive time for sending the questionnaire and the prompts as well as sending highly appealing and respectful reminders to the participants. In addition, this is the first study to address the situation in Sudan.

## Conclusions

In conclusion, this study has shown a relatively lower acceptance rate (55. 8%) for the COVID-19 vaccine and a high vaccine hesitancy state (44. 2%). We found that the main concern of the clinical phase students to defer the vaccine was related to skepticism about the safety and effectiveness of the COVID-19 vaccine corroborated by the widely pervasive negative news. In addition, the role of the University in providing accurate information regarding the pandemic and the vaccine was stated to be unsatisfactory. Hence, we recommend universities in Sudan provide accurate evidence-based information regarding the COVID-19 pandemic and its vaccines safety and efficacy to medical students. More efforts by the Ministry of Health and other governmental bodies are recommended to encourage healthcare trainees to accept the vaccine.

## Supporting information

**S1 File. Raw data of participants.**
(SAV)

## Acknowledgments

We would like to thank all the medical students and their coordinators who participated in the study. Special thanks go to Bilal H. Assamawi who coordinated the recruitment of the medical students.

## Author Contributions

**Conceptualization:** Saud Mohammed Raja, Kabirat Yusuf.

**Data curation:** Saud Mohammed Raja.

**Formal analysis:** Saud Mohammed Raja.

**Investigation:** Saud Mohammed Raja, Kabirat Yusuf.

**Methodology:** Saud Mohammed Raja, Murwan Eissa Osman, Abdelmageed Osman Musa, Kabirat Yusuf.

**Resources:** Saud Mohammed Raja, Murwan Eissa Osman, Abdelmageed Osman Musa, Kabirat Yusuf.

**Supervision:** Murwan Eissa Osman, Abdelmageed Osman Musa, Asim Abdelmoneim Hussien.

**Visualization:** Saud Mohammed Raja.

**Writing – original draft:** Saud Mohammed Raja.

**Writing – review & editing:** Saud Mohammed Raja, Murwan Eissa Osman, Abdelmageed Osman Musa, Asim Abdelmoneim Hussien, Kabirat Yusuf.

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
