## [Decision Letter · Decision Letter 0]

9 Nov 2021

PONE-D-21-29953COVID-19 Vaccine Acceptance, Hesitancy, and Associated Factors among Medical Students in SudanPLOS ONE

Dear Dr. Raja,

Thank you for submitting your manuscript to PLOS ONE. After careful consideration, we feel that it has merit but does not fully meet PLOS ONE’s publication criteria as it currently stands. Therefore, we invite you to submit a revised version of the manuscript that addresses the points raised during the review process.

We look forward to receiving your revised manuscript.

Kind regards,

Sanjay Kumar Singh Patel, Ph.D.

Academic Editor

PLOS ONE

     b) If there are no restrictions, please upload the minimal anonymized data set necessary to replicate your study findings as either Supporting Information files or to a stable, public repository and provide us with the relevant URLs, DOIs, or accession numbers. For a list of acceptable repositories, please see http://journals.plos.org/plosone/s/data-availability#loc-recommended-repositories

Reviewers' comments:

Reviewer's Responses to Questions

**Comments to the Author**

1. Is the manuscript technically sound, and do the data support the conclusions?

Reviewer #1: Yes

Reviewer #2: Yes

2. Has the statistical analysis been performed appropriately and rigorously? 

Reviewer #1: Yes

Reviewer #2: Yes

3. Have the authors made all data underlying the findings in their manuscript fully available?

Reviewer #1: Yes

Reviewer #2: Yes

4. Is the manuscript presented in an intelligible fashion and written in standard English?

Reviewer #1: Yes

Reviewer #2: Yes

5. Review Comments to the Author

Reviewer #1: In this paper entitled “COVID-19 Vaccine acceptance, hesitancy, and associated factors among Medical students in Sudan”, the author investigates the hesitancy of the COVID-19 vaccine and associated factors among medical students in Sudan. A descriptive cross-sectional study was performed using online questionnaires to assess the association between vaccine acceptance and various factors. Although the sample size is small, the study provides knowledge about factors involved in the approval and hesitancy of the COVID-19 vaccine. It would be helpful to develop strategies to remove these hesitancies against the COVID-19 vaccine worldwide.

Minor comments:

1) The statistical technique used in this study is commendable. However, could the authors explain how his results are significant with a small sample size? Also, the study was conducted at faculty that can accommodate more than 1000 medical students. So, what are the reasons primary medical students are not included?

2) The Manuscript concluded that there is a high hesitancy against the COVID-19 vaccine among medical students. However, the medical students are highly educated among whole population. Therefore, could authors explain the reasons behind hesitation among medical students in the discussion, which are different from similar studies mentioned in the manuscript. Also, is this hesitation present in the general population?

4) There are issues in the reference section. Please correct it accordingly.

Reviewer #2: In the current research article entitled " COVID-19 Vaccine Acceptance, Hesitancy, and Associated Factors among Medical Students in Sudan", by Raja et al., have studied/surveyed to estimate determine the acceptance and hesitancy of the COVID-19 vaccine and associated factors among medical students in Sudan. Authors conducted using an online self-administered questionnaire designed on Google Form and sent to randomly-selected medical students via their Telegram accounts from 30th June to 11th July, 2021. Data were analyzed using Statistical Package for Social Sciences software. Chi-square or Fisher's exact test, and logistic regression were used to assess the association between vaccine acceptance and demographic as well as non-demographic factors. They found that, a high level of COVID-19 vaccine hesitancy among medical. This article addresses a research topic of great interest; however, this reviewer has certain suggestions that would help produce a more comprehensive overview of the topic:

Suggestions:

1. What % of COVID-19 vaccine hesitancy is there in Sudan among whole population?

2. The authors may additionally provide one Figure as summary, challenges, or prospect of the present study.

2. The authors should cross-check all abbreviations in the manuscript. Initially, define in full name followed by abbreviation.

3. The English of manuscript can be polished (minor).

4. Authors should add a paragraph to discuss more about the cause of COVID-19 vaccine hesitancy among medical students in Sudan.

---

## [Author Response · Author response to Decision Letter 0]

23 Mar 2022

Reviewers' comments and authors’ reply

Reviewer 1 (Minor comments)

1) The statistical technique used in this study is commendable. However, could the authors explain how his results are significant with a small sample size? Also, the study was conducted at faculty that can accommodate more than 1000 medical students. So, what are the reasons primary medical students are not included? 

Thank you for commending our statistical techniques. Regarding the sample size, the minimum required sample size for the selected study population was calculated with an appropriate formula and the number of participants was successfully attained. Therefore, we believe the sample size is reasonably representative at least to the medical students whom the sample was drawn from. The faculty can indeed accommodate more than 1000 medical students in all their academic years. However, we selected those in their clinical years for the following reasons. First, due to the COVID-19 partial lockdown at the inception of the study, there were interruptions in the academic program and those in their clinical years were thought to be more suitable for inclusion due to their hospital attachments and relatively uninterrupted schedule. Second, the objective of the study was to conduct an online, simple random design survey. Clinical-year students were more active online in their Telegram groups for the sake of coordinating their clinical attachments. Therefore, they were more appropriate for sampling and more likely to meet the inclusion criteria. 

2) The Manuscript concluded that there is a high hesitancy against the COVID-19 vaccine among medical students. 

However, the medical students are highly educated among whole population. Therefore, could authors explain the reasons behind hesitation among medical students in the discussion, which are different from similar studies mentioned in the manuscript. Also, is this hesitation present in the general population? 

A detailed discussion of the reasons behind hesitancy among the medical students has been added in lines 264 through 277. Unfortunately, no published study of COVID-19 hesitancy in the general population exists so far to compare our findings. We have mentioned that in the discussion and recommended such a study in the future.

3) There are issues in the reference section. Please correct it accordingly. 

References have been revised and issues resolved.

Reviewer 2 (Suggestions)

1. What % of COVID-19 vaccine hesitancy is there in Sudan among whole population? 

Unfortunately, no published report currently exists in the literature regarding the percentage of COVID-19 vaccine hesitancy among the general population in Sudan. Investigating the degree of vaccine hesitancy in the whole population was beyond the scope of the study. This has now been mentioned in the discussion section and a recommendation made for a large-scale population study.

2. The authors may additionally provide one Figure as summary, challenges, or prospect of the present study. 

A new figure (Fig 2) has been added as suggested.

2. The authors should cross-check all abbreviations in the manuscript. Initially, define in full name followed by abbreviation. 

Abbreviations have been cross-checked and all issues corrected.

3. The English of manuscript can be polished (minor). 

The manuscript's English has been revised and polished both manually and with the help of the Grammarly app.

4. Authors should add a paragraph to discuss more about the cause of COVID-19 vaccine hesitancy among medical students in Sudan. 

A new paragraph has been added in the discussion section to discuss the cause of COVID-19 vaccine hesitancy in detail (lines 264 - 277).

---

## [Editor Report · Decision Letter 1]

25 Mar 2022

COVID-19 vaccine acceptance, hesitancy, and associated factors among medical students in Sudan

PONE-D-21-29953R1

Dear Dr. Raja,

We’re pleased to inform you that your manuscript has been judged scientifically suitable for publication and will be formally accepted for publication once it meets all outstanding technical requirements.

Kind regards,

Sanjay Kumar Singh Patel, Ph.D.

Academic Editor

PLOS ONE

---

## [Editor Report · Acceptance letter]

30 Mar 2022

PONE-D-21-29953R1 

COVID-19 vaccine acceptance, hesitancy, and associated factors among medical students in Sudan 

Dear Dr. Raja:

I'm pleased to inform you that your manuscript has been deemed suitable for publication in PLOS ONE. Congratulations! Your manuscript is now with our production department. 

Kind regards, 

on behalf of

Dr. Sanjay Kumar Singh Patel 

Academic Editor

PLOS ONE